# The Impact of Thermal Treatment Intensity on Proteins, Fatty Acids, Macro/Micro-Nutrients, Flavor, and Heating Markers of Milk—A Comprehensive Review

**DOI:** 10.3390/ijms25168670

**Published:** 2024-08-08

**Authors:** Yi Wang, Ran Xiao, Shiqi Liu, Pengjie Wang, Yinhua Zhu, Tianjiao Niu, Han Chen

**Affiliations:** 1Food Laboratory of Zhongyuan, China Agricultural University, Beijing 100083, China; wangyi922217@126.com; 2Department of Nutrition and Health, China Agricultural University, Beijing 100083, China; xiaoran881230@163.com (R.X.); liushiqi97@bjfu.edu.cn (S.L.); wpj1019@cau.edu.cn (P.W.); zhuyinhua@cau.edu.cn (Y.Z.)

**Keywords:** direct steam, indirect steam, pasteurization, milk, heating markers, heat-induced gelation, vitamins, minerals

## Abstract

Milk thermal treatment, such as pasteurization, high-temperature short-time processing, and the emerging ultra-short-time processing (<0.5 s), are crucial for ensuring milk safety and extending its shelf life. Milk is a nutritive food matrix with various macro/micro-nutrients and other constituents that are possibly affected by thermal treatment for reasons associated with processing strength. Therefore, understanding the relationship between heating strength and milk quality is vital for the dairy industry. This review summarizes the impact of thermal treatment strength on milk’s nutritional and sensory properties, the synthesizing of the structural integrity and bioavailability of milk proteins, the profile and stability of fatty acids, the retention of macro/micro-nutrients, as well as the overall flavor profile. Additionally, it examines the formation of heat-induced markers, such as Maillard reaction products, lactulose, furosine, and alkaline phosphatase activity, which serve as indicators of heating intensity. Flavor and heating markers are commonly used to assess the quality of pasteurized milk. By examining former studies, we conclude that ultra-short-time-processing-treated milk is comparable to pasteurized milk in terms of specific parameters (such as whey protein behavior, furosine, and ALP contents). This review aims to better summarize how thermal treatments influence the milk matrix, guiding the dairy industry’s development and balancing milk products’ safety and nutritional value.

## 1. Introduction

Milk is comprised of a variety of compounds, including proteins, carbohydrates, lipids, minerals, vitamins, and other newly found bioactive compounds [1]. Due to these compounds, milk is regarded as a natural whole food with a unique flavor and various health benefits, such as antioxidant [2], anti-inflammatory [3], anti-osteoporosis [4] benefits. With consumers’ increasing health awareness, the consumption of dairy products has grown over the past few decades [5]. However, these nutritional compounds are also good substrates for harmful microorganisms, which are not only detrimental to human health but also make the milk susceptible to spoilage [6]. Thus, microorganism control is an indispensable step before commercialization to guarantee milk safety and prolong its shelf life [5].

Of all the methods of microorganism control, thermal processing is the most commonly used, with the advantages of low cost and high efficiency. However, adverse changes occur in milk when exposed to excessive heating conditions, such as the denaturation of proteins and oxidation of lipids, and Maillard reactions are observed in milk after thermal treatment, which further causes some unfavorable flavors, such as cooked, sulfurous, cabbage, and caramelized flavors. Moreover, the loss of heat-sensitive bioactive compounds during heating results in a reduction in the nutritional value of milk [7,8]. In addition, plasmin, a key protease of the fibrinolysis system that regulates sedimentation and age gelation in milk, shows different enzymatic kinetics after various thermal treatments [7]. Research has reported that pasteurization processes, usually referring to heating at 63 °C for 30 min (low temperature long time, LTLT), 72 °C for 15 s, or 85 °C for 2 to 4 s (high temperature short time, HTST), provide minor damage on the flavor, color, bioactive compounds, and rheology properties of milk due to the low processing intensity applied [9,10]. The pasteurization procedure is able to effectively eliminate vegetative microbes but is less feasible against spores and may germinate these aerobic spores in some cases [11], which leads to a short shelf life, requires strict cold-chain transportation, and therefore, limits the supply of milk products with high quality, especially in developing countries [10]. 

Direct steam heating (DSH), a thermal technology introduced to the dairy industry in recent years, is defined as mixing superheated steam directly with the milk matrix to raise the temperature to the targeted value rapidly [12]. This method causes less heat damage compared to indirect heating systems because of the more rapid heating rate and the absence of a heat transfer surface [13]. The processing intensity of DSH is controllable and DSH treatment with a certain temperature and duration [9] might balance the adverse effects of heating on quality and extend the shelf life of milk to produce high-quality dairy products. 

This article briefly summarizes the recent findings of thermal treatments in the dairy industry and compares the impacts of different thermal processes on the structure and content of proteins (whey protein and casein) and lipids, flavor, macro/micro-nutrients (minerals and vitamins), and commonly used heating markers of traditionally pasteurized milk. Afterward, we propose our perspectives on the future optimizations of thermal treatment in milk processing.

## 2. Impact of Thermal Treatment on Structure and Behavior of Main Milk Constituents

Based on heating principles, steam heating treatments can be either composed of an “indirect” system or a “direct” system (Figure 1). In direct heating systems, products are directly mixed with steam under pressure, significantly improving heat transfer efficiency [14]. After this, products are cooled to the targeted temperature, and excess water is removed by vacuum cooling [15]. The rapid heating and cooling processing in the DSH system reduces heat-induced physical and chemical changes [16] compared to indirect steam heating (ISH)-treated milk [17]. Understanding changes in the structure of the main milk constituents, including whey proteins, casein, and milk fat, caused by direct/indirect steam heating, is essential for enhancing its processing stability and controlling its functionalities. Consequently, this section inspects the impact of DSH on the structure and function behavior of the main milk constituents. Meanwhile, the differences in the effects of different heating methods on these substances are also discussed.

### 2.1. Whey Proteins 

Whey proteins, highly ordered proteins in a globular shape, represent the second most abundant protein group (~20% of milk proteins) after caseins. β-Lactoglobulin (β-Lg) and α-lactalbumin (α-La) are the most abundant in whey proteins, which account for approximately half and one-fifth of whey protein, respectively. At the same time, immunoglobulins (IgGs), bovine serum albumin (BSA), and lactoferrin are minor whey proteins. Whey proteins contain several biological functions, such as antioxidant [17], anti-obesity [18], antitumor [19], and immunomodulation [20] functions. Milk needs to undergo heat treatment to guarantee microbiological safety during its shelf life before commercialization. Whey proteins are more sensitive to heating than caseins. HTST pasteurization (73 °C/15 s) showed no significant changes in β-Lg and α-La compared to raw milk, while denaturation of β-Lg and α-La was observed after a 135 °C/2 s steam infusion treatment or a 151 °C/4 s steam injection treatment, which was identified by SDS-PAGE [21,22]. On the other hand, IgGs are the most heat-sensitive among whey proteins, followed by BSA, LF, β-Lg, and α-La [23]. In most cases, ISH treatment with a higher thermal load can significantly denature whey proteins [23,24]. A previous study investigated directly/indirectly heated milk products from 10 plants in six countries and found that denaturation of β-Lg was lower in milk treated with DSH (35–80%) than in milk treated with ISH (79–100%) due to a more rapid heating rate in direct systems [25,26]. In addition to ISH and DSH, the denaturation of whey protein is nearly negligible during pasteurization (72 °C for 15 s) [14]. Wang et al. [27] compared the differences in the thermal damage of proteins (LF, α-La, and β-Lg) in milk treated with different heat treatments, including ISH treatment, direct steam heating treatment (direct steam infusion, DSI, and direct steam injection, DSIJ), and pasteurization. Steam is injected into the food matrix through apertures on the vessel body or through a pipe positioned inside the vessel in DSIJ technology. It involves discharging a series of steam bubbles into the product. Steam injectors are engineered to create a turbulent zone within the steam injector to help mix the steam and the product. DSI technology is used to accelerate steam at speeds of up to 1000 m/s into the food matrix. The steam disrupts the fluid flow and breaks it into small droplets, resulting in an increased contacting surface and higher heat exchange rate. [25,26]. They found that the DSI heating treatment (139–156 °C/0.116–5 s) provided lower thermal damage to α-La, β-Lg, and LF than the ISH treatment, equivalent to pasteurization [27,28,29]. Mi et al. [28] investigated the effects of DSIJ treatment on heat-sensitive serum proteins in yak milk (YM). They found that this heat treatment could reduce the content of LF (about 40%) and β-Lg (about 30%) but had little effect on α-La. However, in contrast, previous studies reported 79–100% β-Lg denaturation, 100% LF denaturation, and 18–54% α-La denaturation after ISH treatment [26,29,30]. Therefore, regarding the denaturation of whey proteins, DSH-treated milk could be more similar to pasteurized milk. Advances in analytical methods and proteomics approaches have revealed the abundance and complexity of whey protein compositions [31,32,33]. Thus, this approach can comprehensively analyze and compare the effects of different heat treatments on the composition of whey proteins in milk. Meanwhile, bioinformatics analysis also helps elucidate these proteins’ potential biological functions. 

Compared to whey proteins with a higher molecular weight (Mw), such as immunoglobulins and bovine serum albumin, there are more complexes formed by the participation of β-Lg and α-La induced by heating, especially β-Lg [14,34]. β-Lg and α-La denature at 75–80 °C and 65 °C, respectively [14]. After an 85 °C/30 min treatment, β-Lg and α-La are irreversibly denatured, exposing the reactive hydrophobic and free cysteine residues of the two whey proteins [35]. They consequently interact with themselves or κ-casein on the surface of the casein micelles in milk through disulfide bonds, thiol/disulfide exchange, and hydrophobic interactions to form whey protein aggregates and β-Lg-κ-casein complexes [36,37]. In the formed aggregates, β-lg interacts with α-La and other β-Lgs [34]. β-Lg unfolds when induced by heating and can start association reactions with other β-Lgs via disulfide bonds. α-La has no free sulfhydryl group, but it associates with β-Lg via thiol/disulfide exchange [23]. The release of reactivated free sulfhydryl groups at temperatures higher than 70 °C induced by the disruption of one of the disulfide bonds triggers and spreads non-reversible aggregation in the same pattern as the natural free sulfhydryl in β-Lg [23]. Yun et al. [38] developed a multistep model for the denaturation of β-lg and its following reaction with κ-casein. They found that the β-Lg denaturation mechanism was expanded to three steps: dimerization (at 63 °C), polymerization (at 80 °C), and reaction with κ-casein (at ~90 °C). In addition, the association of denatured whey protein with the casein micelles depends on pH. At a pH of 6.5, 75–80% of the denatured whey protein was associated with micelles, and this association level decreased to approximately 30% at a pH of 6.7 [39].

### 2.2. Casein

Caseins in cow milk account for approximately 80% of milk proteins, which are present in the form of roughly globular aggregates named micelles with different particle sizes between 50 and 600 nm [40,41]. Casein micelles consist of αS1-casein, αS2-casein, β-casein, and κ-casein (the ratio in cow milk: 4:1:3.5:1.5), colloidal calcium phosphate, and water [42]. Over the last few decades, researchers have proposed different models to illustrate the structure of casein micelles, including the submicelle model [43], the dual binding model [44], and the nanocluster model [41,45]. Although the nanocluster model can best characterize the structural changes of casein micelles in milk caused by different dairy processing methods, the role of β-casein and water is still unclear [45,46]. Therefore, its exact structure in raw milk is still under debate. Dalgleish et al. [41] stated that calcium phosphate nanoclusters linked the αS-caseins and β-caseins; some β-caseins hydrophobically bind to other caseins and stabilize the water channel and pores within the micelles, allowing water, peptides, or other small Mw compounds to pass through the micelles, but these β-caseins can be released from the micelles to the serum phase in milk by cooling [41]. In addition, κ-casein was located on the surface of micelles, providing steric and electrostatic stabilization against aggregation.

Casein micelles resist severe heat treatments because of their lack of a well-defined tertiary structure due to disulfide binding and a large amount of propyl residues, and only temperatures above 120 °C cause the denaturation of casein proteins, which explains the low denaturation rate in pasteurized milk [46]. Upon severe heating, the most critical change in casein micelles is an increase in micellar size due to the association of denatured whey proteins with κ-casein on the surface of micelles [47,48]. The degree of protein particle size change is not the same during different heating conditions. Previous studies found that ISH treatment induces the formation of large-sized aggregates compared to DSH [14,27]. This is because ISH treatment has a higher thermal load than DSH under the same heating conditions. In addition, a previous study compared the effects of two DSH systems (injection and infusion) on milk particle size, and they found that the particle size of steam-infusion-treated milk was lower than that of steam-injection-treated milk [49]. Two possible explanations for this phenomenon are as follows. (a) DSI was gentler. (b) DSIJ induces more self-aggregation of β-Lg, leading to less β-Lg available to associate with κ-casein [9,50,51]. InfusionPlusTM (SPX FLOW, Charlotte, NC, USA) can achieve an extreme temperature (157 °C) in both DSI and DSIJ systems. Thus, our lab used this machine to process skim milk samples over a 139–156 °C temperature range for 0.116–5 s [52]. The results showed that the milk protein particle size was related to the C* value, and it provides a measure of the processing extent of the chemical components of a product [52]. Compared to a low C* value (<0.1; holding times ≤ 0.25 s), milk samples with a high C* value (>0.1; holding times were 3–6 s) had smaller particle sizes, suggesting extensive heat-induced partial dissociation of the β-Lg-κ-casein complex from the casein micelle upon DSI treatment [52]. It should be noted that this complex has been considered the key element that contributes to the formation of age gelation in DSH-treated milk and ISH-treated milk [53,54]. Besides proteins, alterations in salt balance were also observed. DSH induces the calcium precipitation and solubilization of colloidal calcium phosphate, but these changes are invertible at heating temperatures lower than 95 °C for a few minutes [55]. 

The above discussion about casein is based on the milk system, but the impact of DSH on the casein system (without whey proteins, lactose, and salts) needs to be clarified. Therefore, our lab studied the structural changes in casein micelle dispersions upon different DSI treatments (157 °C/0.116 s, 155 °C/3 s, 150 °C/3 s, 145 °C/3 s, and 140 °C/3 s) [44]. As the temperature increased, the particle size of casein micelles reduced, and turbidity decreased significantly [56]. In addition, calcium bridges were deconstructed by DSI and increased electrostatic repulsion inside casein micelles, resulting in the dissociation of casein clusters, and therefore, a looser, smaller, and more porous shape of the casein micelles [56].

### 2.3. Milk Fat

Fat in milk is present in the form of fat globules. The core of milk fat globules is triacylglycerols (TAGs), which are surrounded by three-layered biofilms called milk fat globule membranes (MFGMs) [57]. These membranes consist of some membrane-specific proteins, triglycerides, cholesterol, phospholipids, and enzymes [34]. 

MFGM protein occupies 1–2% of total dairy protein, which is much lower than casein and whey protein [34,58]. The major MFGM proteins include fatty acid-binding protein (FABP), xanthine oxidase (XO), cluster of differentiation 36 (CD36), periodic acid Schiff glycoprotein III (PAS III), adipophilin (ADPH), butyrophilin (BTN), periodic acid Schiff glycoprotein 6/7 (PAS 6/7), and Mucin-1 (MUC 1) [58]. These proteins are presented in different layers of MFGMs [59]. ADPH appears in the inner monolayer, while XO is located in between both layers [15]. MUC 1, PAS III, CD 36, and BTN are in the outer layer [15]. PAS 6/7 is only loosely bound to the outer layer of MFGMs [15]. Thermal treatment at >65 °C accelerates the binding of the denatured whey proteins to MFGM proteins, as well as the dissociation of natural MFGM proteins. The denatured β-Lg can undergo complexation with α-La, other β-Lgs, and minor whey proteins with a higher Mw in milk [59]. These complexes can interact with MFGM proteins mainly through β-Lg as a “connector” [59]. In addition, α-La, IgGs, LF, and BSA also directly bind to MFGM proteins [34]. MFGM protein dissociation is not observed in pasteurized milk and the release of BTN (at >85 °C, 3 min), XO (at >85 °C, 3 min), as well as PAS 6/7 (at >80 °C, 3 min) can be observed above certain temperatures [60,61]. According to the SDS-PAGE patterns regarding major MFGM proteins in milk treated with different heat treatments (pasteurization, DSH, and ISH), we found that pasteurization and DSH retain more major MFGM proteins. However, the significant denaturation of major MFGM proteins in milk treated with ISH treatment is observed (up to 50.05% MFGM denatured after a 135 °C/5 s treatment) because of the higher thermal load exerted by ISH treatment [15,34,62]. The condensation and collapse of steam bubbles during DSIJ lead to homogenization and cavitation effects. In addition to these changes, it should be noticed that the homogenization effect can influence the reconstruction of the MFGM [14].

The polar lipids on the MFGM account for 0.1–0.3% of total milk lipids, which can be divided into glycerophospholipids (phosphatidylcholine, PC; phosphatidylinositol, PI; phosphatidylserine, PS; and phosphatidylethanolamine, PE) and sphingolipids (mainly sphingomyelin, SM) [59]. Polar lipids endow the MFGM with fluidity, but the distribution of polar lipids is non-homogeneous. Liquid-ordered domains (SM+cholesterol) with low fluidity lead to a more fragile membrane structure, which can affect the properties of fat globules. Heat treatment causes the membrane to become “brittle”. Huang et al. [63] investigated the effects of different pasteurization treatments on the physical properties and interfacial composition of bovine milk fat globules. For polar lipids, they found that the phospholipid layer of the MFGM was incomplete upon heating [63]. Meanwhile, the liquid-ordered domains were more easily shed into the aqueous phase. As a result, the nano-mechanical properties of the MFGM decreased [63]. However, the effect of DSH treatment on MFGM phospholipids is still unclear.

## 3. Impact of Thermal Treatment on Flavor of Milk

It has been reported that high processing temperatures influence interactions between dairy compounds in the milk matrix or with the environment, such as the Maillard reaction between proteins and carbohydrates [64,65], lipid oxidation [66], and protein degradation [67]. The results of these heat-induced reactions are eventually reflected in the flavor of milk. 

Zhao et al. [68] reported that traditional pasteurized milk (63 °C/30 min or 72 °C/15 s) has a higher content of low Mw constituents, such as 2-butanone and dimethyl ketone, while ultra-high temperature treatment with DSI (150 °C, 0.1 s) increases the content of ketones and aldehydes with a higher Mw, such as hexanal, pentanal, heptanal, benzaldehyde, and nonanal. Of these, sulfur-containing compounds such as dimethyl sulfone are related to the cooked taste, hexanal is a compound that has a grassy off flavor, and benzaldehyde is an aromatic substance that emanates a pleasant flavor formed by the Maillard reaction [69]; nonanal is a common flavor substance which emits floral and citrus aromas [70]. UHT milk normally has stronger flavors than HTST milk, especially regarding the cooked flavor and a sulfur or eggy flavor, which are not liked by US consumers who typically prefer light or medium milk flavor intensity [71]. This bias can be dependent on the type of milk most commonly consumed. Liem et al. [72] reported that consumers from China, where 60% of the milk consumed is long-life milk (LLM), preferred the flavor of LLM, whereas consumers from Australia, where only 10% of milk consumed is LLM, preferred HTST milk.

Meng et al. [73] comprehensively compared both DSI and DSIJ treatment (140–155 °C, 0.5–2 s) to HTST treatment in terms of the flavors of milk obtained with two extraction methods, namely solvent-assisted flavor evaporation (SAFE) and solid-phase microextraction (SPME). There were 59 volatile compounds discovered in DSI milk and DSIJ milk. There were 19 and 54 substances detected by SPME and SAFE, respectively, of which 14 substances were detected by both extraction methods. The 14 compounds include hexanal, nonanal, 2-undecanione, δ-decanolactone, dimethyl sulfone, and some short-chain acids. SAFE is effective for the extraction of alcohols, fatty acids, and aldehydes as reported, while SPME is better for the extraction of esters and sulfur-containing compounds [74]. 

The number of volatile compounds in DSI milk is a bit less than in DSIJ milk (50 vs. 52 substances). Flavor compounds, such as dimethyl sulfone, nonanal, 2-decanone, and 3-hydroxy-2-butanone, were detected in DSIJ milk but not in DSI milk. In terms of content, combining the results of two extraction methods, the content of 2-undecanone and decanoic acid in DSI milk is higher, while the content of dimethyl sulfone is lower. 

## 4. Impact of Thermal Treatment on Macro/Micro-Nutrients in Milk

### 4.1. Minerals 

It is well known that temperature fluctuations can change the distribution of minerals between the colloidal and aqueous phases [75]. The processing conditions, such as temperature, dilution ratio, pH, and the presence of chelating agents or salts, lead to an imbalance of the minerals in milk [76]. The increase in temperature facilitates the disassembly of the inorganic phosphate group (Pi) in the aqueous phase and breaks the balance between H_2_PO^4−^ and HPO4^2−^ in milk. Calcium phosphate becomes supersaturated in the aqueous phase with lower concentrations of ionic calcium and Pi. The changes in mineral balance depend on the intensity of the heat treatment between the colloidal and aqueous phases [75,77]. The HPO_4_^2−^ concentration is more affinitive for calcium, and it reacts at high temperatures with calcium to form calcium phosphate salts. Heat treatment causes the aggregation of calcium phosphate because its solubility decreases at higher temperatures. The content of P and Ca in serum decreases by 35% and 60%, respectively, after being heated from 20 to 90 °C. Moreover, magnesium concentrations in the colloidal phase increase from 1.53 to 1.97 mM but there is little change in potassium and sodium [78,79]. Pouliot et al. [80] also reported that the recovery ability of the heat-induced (85 °C/40 min) changes in P and Ca was 93–99% and 90–95%, respectively. 

Boiani et al. [81] reported that Pi and Ca within micellar casein played vital roles as inorganic calcium phosphate salts or in forming casein phosphate nanoclusters at low concentrations of micellar casein during a low heating strength (25 to 60 °C). With more intensive heating (60 to 80 °C), the higher temperature fortified the negative charge of Pi, leading to an interaction between micellar casein and the casein phosphate nanocluster. It has been reported that treatment at 100 °C for 15 min molded a new type of calcium phosphate with the nature of an alkaline salt other than the native colloidal calcium phosphate in fresh milk, while the structure of the freshly formed calcium phosphate due to treatments at below 90 °C for 15 min was identical with native colloidal calcium phosphate nanoclusters [82]. 

The alterations in milk’s mineral balance triggered by thermal treatment are of great importance to the dairy industry [83]. It has been reported by Jeurnink et al. [84] that up to 45% of minerals, including 15.7% calcium, presented in the residues on the heater of a plate heat exchanger with an operating temperature range of 69–85 °C. Moreover, only 0.2% of colloidal minerals in milk from the aqueous phase were adhered to the heater surface after treatment at 85 °C for 1 min. The shifting balance of the minerals results in the poor coagulation of the milk during cheese production [84]. Additionally, severe treatment (115 °C/ 40 s) could cause some extent of mineral losses in milk because of the fouling mechanism related to the severity of thermal treatment. 

For the bioavailability of minerals, Seiquer et al. [85] found that overheating (three cycles of 116 °C/16 min heating) decreased the amount of soluble calcium by about 23.6% in milk; which was its bioaccessibility after an in vitro digestion compared to that of 150 °C/6 s treatment. The reduction in solubility of Ca could be due to dephosphorylation triggered by the thermal treatment of casein phosphopeptides, which prevents it from aggregation and the formation of chelates between calcium and Maillard reaction products. No significant effect of ultra-heating on the absorption of Ca was also determined using a rat model, in which food intake was administrated.

### 4.2. Vitamins 

Vitamin B_1_ (thiamin), is normally abundant in products from animal sources in the form of phosphorylation and is likely to be denatured by oxidizing, reducing agents, and thermal conditions (as seen in Table 1) [86,87] The circumstances of the thermal treatment module, e.g., plate or tubular energy exchanger in thermal treatment and DSI or DSIJ treatments, showed limited effects on vitamin B_1_ levels in milk [88]. Contrarily, for in-bottle sterilized milk, rare in the dairy industry nowadays, the loss of vitamin B_1_ might be up to 40% [89]. No significant decrease in vitamin B_1_ is observed when opened LLM milk was refrigerated for 10 days [90]. Conversely, as for evaporated milk, the vitamin B_1_ loss can be up to 50% [91].

Vitamin B_2_ (riboflavin) in milk is insensitive to thermal treatment [92,93], but its stability is weakened when exposed simultaneously to light [94]. It has been reported that skim milk treated from 80 to 120 °C showed a declining trend in photolysis of vitamin B_2_, possibly due to the denaturation of whey protein and/or an increase in the size of casein micelle, which blocks light [95]. Aminoreductone, a kind of Maillard-reaction product, is the primary compound promoting the stability of vitamin B_2_ against light in LLM [96]. Vitamins B_3_ (niacin), B_5_ (pantothenic acid), and B_7_ (biotin) in milk were insensitive against prolonged pasteurization below 72 °C and following storage at 4 °C [93,97]. A reduction of less than 10% in vitamins B_3_, B_5_, and B_7_ in pasteurized milk was also reported in former studies [91,98,99].

**Table 1 ijms-25-08670-t001:** Influences of different thermal treatments on vitamins in milk.

Vitamins		Heating Conditions (Whole Milk Unless Stated Otherwise)	Raw Milk	After Treatment	Ref.
water-soluble vitamins	Vitamin B_1_ (thiamin) (μg/mL)	77 °C/15 s	0.45 ± 0.02	No loss	[87]
		94 °C/420 s	0.43 ± 0.03	6.34% loss	
		129 °C/2 s, indirect	0.44 ± 0.02	No loss	
		140 °C/4 s, direct	0.46 ± 0.03	No loss	
		141 °C/4 s, indirect	0.50 ± 0.03	No loss	
	Vitamin B_2_ (riboflavin)	77 °C/15 s	1.42 ± 0.01 μg/mL	No loss	[87]
		94 °C/420 s	1.43 ± 0.01 μg/mL	3.68% loss	
		129 °C/2 s, indirect	1.39 ± 0.01 μg/mL	1.80% loss	
		140 °C/4 s, direct	1.42 ± 0.01 μg/mL	1.41% loss	
		141 °C/4 s, indirect	1.35 ± 0.01 μg/mL	No loss	
		75 °C/6 s	0.17 ± 0.01 μmol/L	5.9% loss	[98]
		85 °C/6 s	0.17 ± 0.01 μmol/L	No loss	
		92 °C/6 s	0.17 ± 0.01 μmol/L	No loss	
	Vitamin B_3_ (niacin)	62.5 °C/30 min	218 μg/100 g	No loss	[97]
		75 °C/15 min	238 μg/100 g	No loss	
		75 °C/6 s	13.28 ± 0.44 μmol/L	No loss	[98]
		85 °C/6 s	13.28 ± 0.44 μmol/L	No loss	
		92 °C/6 s	13.28 ± 0.44 μmol/L	No loss	
	Vitamin B_5_ (pantothenic acid) (μg/100 g)	62.5 °C/30 min	469	No loss	[97]
		75 °C/15 min	610	4.8% loss	
	Vitamin B_6_ (pyridoxamine) (μmol/L)	75 °C/6 s	2.06 ± 0.60	No loss	[98]
		85 °C/6 s	2.06 ± 0.60	No loss	
		92 °C/6 s	2.06 ± 0.60	No loss	
	Vitamin B_7_ (biotin) (μg//100 g)	62.5 °C/30 min	0.73	No loss	[97]
		75 °C/15 min	0.95	No loss	
	Vitamin B_9_ (folic acid) (μmol/L)	75 °C/6 s	0.75 ± 0.06	No loss	[98]
		85 °C/6 s	0.75 ± 0.06	No loss	
		92 °C/6 s	0.75 ± 0.06	No loss	
	Vitamin B_12_ (cyanocobalamin) (ng/mL)	77 °C/15 s	3.60 ± 0.40	No loss	[87]
		94 °C/420 s	3.30 ± 0.20	23.30% loss	
		129 °C/2 s, indirect	3.40 ± 0.60	No loss	
		140 °C/4 s, direct	4.40 ± 0.30	No loss	
		141 °C/4 s, indirect	3.30 ± 0.10	No loss	
		100 °C/1 h	100 μM (standardized)	10.2% loss	[100]
		100 °C/1 h (2% fat)	100 μM (standardized)	12.5% loss	
		100 °C/1 h (fat-free)	100 μM (standardized)	14.4% loss	
		76 °C/16 s	0.31 μg 100 g^−1^	No loss	[101]
		96 °C/5 min	0.31 μg 100 g^−1^	3.2% loss	
	Vitamin C (ascorbic acid) (μg/mL)	50 & 63 °C/10–60 min	100	~15% loss at 10 min and up to 40% loss at 60 min	[99]
		75 & 90 °C/0.25–10 min	100	~12% loss at 0.25 min and up to ~35% loss at 10 min	
fat-soluble vitamins	Vitamin A (retinol) (IU/L)	63 °C/30 min	325 ± 19.2	2% loss	[102]
		121 °C/15 min	325 ± 19.2	36.6% loss	
	Vitamin D_2_ (ergocalciferol) (IU/L)	63 °C/30 min	594.28 ± 2.22 (fortified)	No loss	[103]
		121 °C/15 min	594.28 ± 2.22 (fortified)	No loss	
	Vitamin E (tocopherol) (μg/mL)	77 °C/15 s	0.97 ± 0.03	No loss	[87]
		94 °C/420 s	0.96 ± 0.03	No loss	
		129 °C/2 s, indirect	0.94 ± 0.04	No loss	
		140 °C/4 s, direct	1.00 ± 0.06	No loss	
		141 °C/4 s, indirect	0.98 ± 0.04	No loss	

The effect of a reduction in vitamin B_6_ (pyridoxamine) in pasteurized (<8%), evaporated (35–50%), sterilized (20–50%), and LLM (<10%) milk after storage on shelf-life duration was addressed [104]. Interconversion of different forms of vitamin B_6_ may occur because of heat treatment, being more pronounced in evaporated milk. A more recent study reported that thermal treatment of 120 °C/400 s can alter the distribution of vitamin B_6_ forms. It proposed this as a discriminator of high-temperature treated milk for a novel integrator for time–temperature combination from other thermally processed counterparts [105].

Vitamin B_9_ (folic acid) (5-methyltetrahydrofolate) is presented as part of folate-binding proteins (FBPs) in fresh milk [106]. FBPs are partly denatured in pasteurized milk and extensively denatured in exceeded-heated milk. Thus, folate presents as free folate in ultra-heated milk. It remains unclear how FBPs affect the bioavailability of folate. It was earlier reported that FBPs play a role in the direct transport of folate via intestinal mucosa. However, studies published more recently showed that FBPs were less effective in the promotion of the bioavailability of folate [107].

Vitamin B_12_ (cyanocobalamin) shows good stability against pasteurization and ultra-high-temperature treatments [108], but thermal treatment of milk at a higher strength may cause up to a 20% loss in vitamin B_12_ [91]. Despite vitamin B_12_ being insensitive to heat in plain pasteurized or ultra-heated milk, a remarkable loss (~1/3) was observed in chocolate milk processed at 100 °C for 1 h compared to no-chocolate-added milk (<10%), as reported by Johns et al. [100]. These results may be due to the fact that cocoa powder is physically affinitive to vitamin B_12_ and/or has a high capacity for peroxide generation of cocoa polyphenols (i.e., (+)-catechin) and related products such as caffeic acid, (−)-epigallocatechin, and gallic acid. Vitamin B_12_ loss is inappreciable in milk treated at 96 °C for 5 min [101] to 100% in ultra-heated milk stored for 20 weeks at room temperature [109]. Heat treatment might not be a significant factor in the instability of vitamin B_12_. However, the presence of oxygen-sensitive ingredients such as copper and ascorbic acid as well as oxygen exposure accelerate the destabilization of vitamin B_12_ during thermal treatment.

Various factors could alter the capacity of vitamin C (ascorbic acid) against thermal treatment controlled by a group of factors including thermal treatment patterns, de-airing processes as well as storage conditions and periods. The loss of vitamin C in pasteurized cow milk refrigerated at 4 °C for 6–9 d may be up to 45% [110,111]. As the temperature increases from 63 °C to 100 °C for 30 min, the vitamin C losses in cow milk and camel milk (CM) increase from 18 to 48% and 27 to 67%, respectively [112]. Contrarily, the loss of vitamin C is inappreciable in LLM milk when milk was kept at low oxygen concentrations (<3.3 ppm) [111]. The food matrix considerably affects the rate constant of vitamin C degradation and a higher rate constant for the denaturation of vitamin C is observed in milk than in polyphenol-rich drinks. The duration of thermal treatment and storage, the oxygen concentration, and the presence of antioxidants are essential for the stability of vitamin C [113].

In light and air, there may be a loss of vitamin A (retinol) with a prolonged holding time during thermal treatment [114] or the storage of processed milk at ambient temperatures [115]. Fortified vitamin A is stable against pasteurization but remarkably decreases up to 23% and 37% in boiled and sterilized milk, respectively [102]. Additionally, the fortification of milk with ferrous gluconate hydrate or ferric pyrophosphate soluble adversely affects the thermal stability of vitamin A. The evaporation of pasteurized milk may also cause a reduction in vitamin A, but not carotene [116,117]. Vitamin D (ergocalciferol) is more sensitive to light degradation than thermal treatment, similar to vitamin A [118,119]. Contrarily, the presence of oxygen shows little effect on the stability of milk fortified with vitamin D_3_ (cholecalciferol) [103].

The pasteurization or evaporation process shows less effect on Vitamin E (tocopherol) degradation [116,120]. Moreover, minor damage of either δ- or γ-tocotrienol in LLM after storage was observed [120]. A partial degradation of vitamin E (especially α-tocopherol) is demonstrated when exposed to air, being more profound in unesterified tocopherols due to free phenolic hydroxyl groups [114,121]. Milk proteins protect vitamin E against heat in dairy products, like vitamins A and D. Vitamin K in milk is unlikely to be affected by heat treatments, just like vitamin E [104].

Overall, thermal treatment widely utilized in the dairy industry (i.e., HTST, ESL or LLM milk, direct/indirect heat treatment) shows limited effects on the denaturation of milk vitamins, except for vitamin C, which is more sensitive to heating in water-soluble vitamins. Usually, milk is not preferable to supplement water-soluble vitamins except for vitamin B_2_ and vitamin B_12_, and there is minor damage to most vitamins caused by thermal treatment. Interactions between vitamins and other milk components, such as casein and lipids, triggered by heat might be a protective barrier, promoting the thermal stability of vitamins in milk.

## 5. Influences of Heat Treatment on Heating Markers in Milk

### 5.1. Furosine and Lactulose

The Maillard reaction is commonly seen during food processing when heating is involved, and it generates multiple products, including precursor substances of furosine, which can cause kidney damage [122]. Furosine has been observed to have a higher level with higher dissolved air/oxygen contents, possibly due to reactions on the oxidative side [123]. Elliott et al. [124] compared heat treatment directly or indirectly (147 °C/6 s) and found that the DSH process achieved a lower level of furosine by 76.6% on week 1 and 27.9% on week 24. 

Furosine is typically less in direct-heated milk than in indirect-heated counterparts for 50–70% [125,126,127]. Lee et al. [128] demonstrated that ultra-pasteurization treatment accelerated the generation of furosine. However, DSI samples had a much lower content than that of indirect heat treatment samples (43.81 mg vs. 168.72 mg/100 g protein) while the HTST samples had the lowest content of 6.95 mg/100 g protein. Furosine formation was only remarkably influenced by the extreme processing conditions (i.e., holding for 60 s at temperatures > 100 °C) when pasteurized either directly or indirectly [127]. Furosine levels for raw and pasteurized milk are 4–5 mg and 4–7 mg/100g protein, respectively, while furosine levels for milk processed under extreme conditions could be up to 372 mg/100 g protein (150 °C/20 s). 

As for extreme short-time DSH (<0.5 s), Rauh et al. [129] compared two DSH methods with different preheating treatments (95 °C/5 s or 95 °C/180 s followed by 150 °C/0.2 s) to ISH (140 °C/4 s). They found that both pretreatments followed by 150 °C/0.2 s treatments generated dramatically less furosine than 4 s indirect heating (11.6 mg/100 g protein or 7.6 mg/100 g protein vs. 119.9 mg /100 g) of protein. Similar results were reported in a study on DSIJ treatment with an extremely short timeframe (154~156 °C/0.116 s), and only a slight influence was found in the level of furosine in YM (~11.1 mg for processed milk vs. ~8.7 mg for raw milk) [28,130].

Lactulose is an isomer of lactose that is formed during the thermal processing of milk, also used as a marker for the thermal treatment of milk. Marconi, E. et al. [131] conducted a comprehensive study testing lactulose content in milk samples after different thermal treatments, labeled as follows: pasteurized (PAST) milk, high-temperature pasteurized (HT PAST) milk, direct UHT-treated milk using an injection system (INJ UHT), direct UHT-treated milk using an infusion system (INF UHT), indirect UHT-treated milk using a plate or a tubular heat-exchange system (IND UHT), and in-container sterilized (STER) milk, respectively. The content of lactulose is closely related to thermal treatment intensity as follows: 744 mg/L for STER milk, 348 mg/L for IND UHT milk, 165 mg/L for INJ UHT milk, 107 mg/L for INF UHT milk, 58 mg/L for HT PAST milk, and 3.5 mg/L for PAST milk (the European Union limit for milk is 600 mg/L). 

### 5.2. Alkaline Phosphatase (ALP)

Alkaline phosphatase (ALP) inactivation is commonly regarded as a maker of sufficient pasteurization, and its inactivation is well known to occur even for short-time pasteurization of at least 70 °C for 16 s [132]. Therefore, ALP activities in milk samples are regarded as a marker of heat treatment severity [133]. 

Dickow et al. [134] measured changes in ALP activity with heat treatment of 63 °C/ 30 s, 72 °C/15 s, or 150 °C/0.1 s and found that ALP was completely inactivated when heated at 72 °C or above for all three methods. Only from the perspective of ALP activity, thermal treatment at or above 72 °C can be regarded as standard HTST treatment, which agrees with Fox and Kelly [132]. Lorenzen et al. [88] compared ESL milk samples, heated indirectly or directly, to HTST and ultra-heated milk samples regarding indigenous milk enzymes. The activities of alkaline phosphatase (ALP), lipase (LIP), and lactoperoxidase (LPO) were described. The results of LPO proved that a sufficient thermal treatment was applied to directly and indirectly heated ESL milk along with ultra-heated milk as LPO activities of all samples were below 5 U/L, which is 300–2100 U/L for HTST milk samples. As for ALP and LIP activities, all samples were in a similar range (0.01–0.50 U/L).

### 5.3. Lactoperoxidase (LPO)

Lactoperoxidase (LPO), one of the most abundant enzymes in bovine milk, exerts its function as a nature-sourced antibacterial agent, and it has been reported that lactoperoxidase can alleviate symptoms of asthma and reduce the damaging effects of hydrogen peroxide [133,134]. LPO activity can be used as an indicator of whether the milk was heated over 78 °C for 15 s [135]. It has been reported that the LPO activity of fresh milk decreased by 58% and 92% in milk after being heated at 72.5 °C for 15 s and 25 s, respectively [136]. This finding was reviewed by Lan et al. [137], with the LPO activity dramatically reducing after heating at 75 °C for 15 s and completely deactivating after heating at 85 °C or above for 15 s.

### 5.4. Plasmin

Treatment with a low temperature (63 °C/30 min) significantly augments plasmin activity in bovine casein fractions. The results of plasminogen are similar to plasmin; 63 °C/30 min treatment has similar plasminogen-derived activity compared to its raw counterpart, but an extended time heating treatment reduces plasminogen-derived activity up to 100%. Similar results were reported by van Asselt et al. [138] that heat treatment of 80 °C/300 s with a 45 s pre/post-treatment could eliminate up to 99.96% plasmin activity. Leite et al. [139] reported that extended-time heating (85 °C/5 min) treatment could cause a significant decline (~100%) of plasmin activity in both whey and casein fractions of bovine milk. Ultra-high temperature treatment exhibits lower efficiency on plasmin inactivation; 135 °C /4 s treatment could decrease plasmin activity by about 70% compared to raw milk [140]. An other study by Leite et al. [141] described the impact of conventional pasteurization (63 °C/30 min or 75 °C/15 s) on cathepsin D and elastase activities. Both heat treatments decreased the activity of cathepsin D by ~45%. As for elastase, 63 °C/30 min treatment showed no effect, and 75 °C/15 s treatment reduced elastase activity by about 25%. 

## 6. Impact of Heat Treatment on Other Properties of Milk

### 6.1. Age Gelation

Ultra-instantaneous direct steam heating (UI-DSH) treatment has emerged based on the heating system of steam infusion, which can achieve more extreme heating conditions (>155 °C for <0.1 s) [15]. Compared to ISH-treated whole milk, the levels of bio-active proteins and flavor in UI-DSH whole milk are similar to pasteurized whole milk [15] (Figure 2). However, some detrimental changes, including bitterness, creaming, age gelation, and sedimentation/clarification, can occur [53]. Of these, creaming is the first observed defect, seriously affecting the quality of milk and consumers’ purchase intentions.

To remain stable for at least 6 months, commercial ISH-treated whole milk undergoes sufficient homogenization; however, an obvious creaming can occur in UI-DSH milk in a short time, because of residual plasmin attacks on interfacial proteins of fat globules. Our previous study determined that the activity of residual plasmin was 1.25 mU/mL in commercial UI-DSH whole milk [54]. Unfortunately, residual plasminogen was gradually converted to plasmin by an activator during storage [55]. Thus, at 25 °C and 37 °C, plasmin activity increased rapidly within 15 days and reached a maximum value (7–8 mU/mL) between days 15 and 45 [142]. 

The natural MFGM is a three-layer structure; membrane proteins are either integrally or peripherally attached to the membrane [59]. After homogenization disrupts the membrane, caseins are involved in reconstructing the fat globule interface, forming a mixed of three-layer and monolayer structure [34]. This complex structure affects the hydrolysis behaviors of plasmin. In this case, our lab revealed the mechanisms underlying fat destabilization in UI-DSH whole milk at the molecular level induced by added plasmin, and further investigated the impacts of the MFGM structure on the hydrolysis of major MFGM proteins. The results showed that plasmin decreased the zeta-potential value as well as the amount and coverage of interfacial proteins through hydrolysis, causing electrostatic and steric destabilization of fat globules, and thus the fat globules flocculated and coalesced. It is worth noticing that the co-hydrolysis of membrane-anchoring proteins (X0-BTN-ADPH) and caseins was the direct cause of fat globule destabilization [15]. Moreover, at the interface, caseins, BTN, and ADPH were very sensitive to plasmin, while it was difficult for PAS 7 to be hydrolyzed by plasmin [15]. Overall, the complex interfacial structure reduced the susceptibility of some major MFGM proteins to plasmin and provided protective effects [15].

Three methods should be considered to minimize creaming in whole UI-DSH milk during storage, including plasmin activity and the modulation of the components of interfacial proteins [15]. Plasmin activity causes fat destabilization directly in whole UI-DSH milk. Minimizing this issue should involve making a trade-off among some factors, including thermal load, residual plasmin activity, and the retention of bioactive components. Although increasing homogenization pressure within a specific range can reduce the particle size of fat globules, it will provide more caseins at the newly created water–oil interface. Thus, the modulation of the constitution of interfacial proteins will be a new direction to minimize the rise of fat; namely, the proteins involved in the reconstruction of the fat globule membrane are replaced with milk proteins that are resistant to plasmin.

### 6.2. Effect of Thermal Treatment on Unconventional Milk

Unconventional milk lactated by minor mammalian species, such as goats, yaks, and camels, is often accompanied by unique nutritional, textural and medicinal virtues. Ultra-high temperature processing and pasteurization are widely utilized to guarantee milk quality and safety. Goat milk (GM) has been gaining attention recently due to its unique nutritional value, digestive virtues, and less allergenicity than cow milk due to low β-lactoglobulin and αs1-casein content [143]. Moreover, the easy digestibility of GM fat is possibly attributed to its smaller diameter of 3.2–3.6 μm and its higher concentration of short to medium-chain fatty acids, which are used to treat cholesterol disorders, anemia, metabolic disorders, and low mineral bone density [143]. YM is economically essential for herders in the Chinese Qinghai–Tibet Plateau, with 1.2 million tons of annual production. Additionally, YM has different nutritional value than cow milk since the contents of protein, fat, and dry matter in YM are ~5, ~6 g, and ~17 g/100 mL, respectively [144,145], so there is an excellent opportunity for high quality, unique YM products. CM exerts various medicinal and therapeutic values, including anti-cancerous, anti-diabetic, hypo-allergenic, ACE-inhibitory, and cholesterol-lowering properties [146]. These unique features of unconventional milk lead to certain limitations related to intensive thermal treatments, such as nutritional losses, off-flavors, and color modifications. 

GM is comparatively sensitive to thermal treatment compared to bovine milk due to its lower concentration of citrates and micellar solvation but its higher concentration in ionic calcium, making it more fragile against heat [147]. Heilig et al. [148] compared GM to bovine milk after heating using different temperature–time combinations (72 °C, 32 s; 120 °C, 4 s (DSIJ/standard tubular)/10 s (DSI); 140 °C, 4 s (DSIJ/standard tubular)/10 s (DSI)). The sensory quality (descriptive sensory analysis and triangle test), casein micelle diameter (dynamic light scattering), and storage stability (precipitation at 4 and 20 °C) of differently treated milk were analyzed. While GM heated at 72 °C showed neither processing nor sensory deficiencies, GMs heated at 120 and 140 °C were observed to be slightly to more clearly grainy and precipitated after 21–90 days of storage. As for storage stability, sensory quality, and process suitability, skim milk showed overall better results than whole milk, and DSH techniques gave better overall results than indirect ones. They also reported in-process coagulation due to casein micelle aggregation in the case of indirectly heated GM.

A study focused on the digestive kinetics of YM after pasteurization (65 °C, 30 min), autoclaving (121 °C, 20 min), microwaving (2.45 GHz at 640 W, 140 s), and non-heated treatment (raw milk) using an in vitro model [149]. Due to the formation of clots with denser structures, a shorter gastric emptying half-time for the pasteurized (36.2 min) and microwaved (34.2 min) milk was determined compared to that of the autoclaved (41.1 min) and fresh (38.4 min) milk. The highest protein digestibility was found in pasteurized milk (92.5%) at the end of digestion, followed by milk treated by microwaving (87.8%), autoclaving (86.1%), and raw milk (80.8%). Mi et al. [28] comprehensively studied the influence of DSI treatment on YM with temperatures of 138 (5 s)–156 °C (0.116 s). They found that DSI treatment effectively sterilized *G. stearothermophilus* spores without extensively degrading serum proteins. Serum protein denaturation in milk caused by DSI was lower than that normally found in traditional ultra-high-temperature treated milk, with losses of 30 and 40% of native β-Lg and lactoferrin, respectively, but not for α-La, which was more similar to pasteurization. Moreover, the heating makers, such as lactulose and furosine in DSI milk, were both below the limits for pasteurized milk (50 mg L^−1^ and 0.12 mg g^−1^ of protein), which indicates the potential that DSI-treated milk might be marketable as a pasteurized product.

Understanding milk’s heat stability is important to the dairy industry since thermal processing is commonly used. The coagulation time caused by thermal treatment (CTT) of CM at 130, 120, and 100 °C was investigated in the pH range of 6.3–7.1 [150]. At 130 and 120 °C, the milk was unstable in a wide pH range, with a CTT of 2–3 min. The CTT initially increased to 12 min at 100 °C, then remained constant in a pH range between 6.4 and 6.7 and increased significantly with the increasing pH until approximately 33 min was reached at a pH of 7.1. CM seemed much less stable than bovine milk during heat processing [151]. Similar results were reported by Sagar et al. [152]; the CTTs at 140 °C for CM, buffalo, and bovine milk were determined to be 133.6, 1574.6, and 1807.4 s, respectively. The presence of κ -casein and β-lactoglobulin and the interaction between them during heating is critical for retaining milk stability [150]. Therefore, the lower level of κ-casein (5% of casein in CM vs. 13.6% of casein in bovine milk) and the shortage of β-lactoglobulin may explain CM’s instability at high temperatures. Genene et al. [153] compared the denaturation of whey protein and the resultant rennetability of CM to those of bovine milk under heat treatment from 65 °C to 90 °C with different durations, and found that α-La showed less denaturation in CM than bovine milk. β-Lg was not found in CM, which could be one of the main differences compared to bovine milk, while in bovine milk, both β-Lg and α-La denatured along with increasing heat treatment. The gelation properties of CM were significantly affected by the severity of the thermal treatment applied. Zhao et al. [154] analyzed the changes in products regarding the Millard reaction after fresh CM was heated at different temperature–time combinations as well as volatile components. 

Markers in CM showed different changes compared to bovine milk. A former study showed that camel ALP is heat-resistant, still showing activity at 90 °C [155], which is typically inactivated at around 72 °C [103]. On the other hand, Tayefi-Nasrabadi et al. [156] determined that camel LPO was more heat-sensitive than bovine LPO. There are insufficient in-depth studies on convenient markers of pasteurization for CM, which limits the establishment of an international standard. Thus, the pasteurization of CM may be incorrect and needs to be revised. Heat-induced changes in milk could affect the properties of reconstituted powders for applications in end-products. 

## 7. Conclusions

Milk is rich in nutrition and is an easy-to-obtain food material; therefore, it is vital to explore its thermal behavior, texturally and nutritively. Although micronutrients and bioactive compounds are normally present at low levels, they present massive influences on the health of humans, especially infants. Developing more sensitive and precise analysis techniques has improved our understanding of the importance and functions of these minor components. Thermal treatment is an essential tool for the dairy industry to ensure microbiological safety and maintain milk quality. On the other hand, heating can cause a loss of macro/micro-nutrients by decomposition or inducing oxidation.

However, pasteurization (<85 °C/ 60 s) has a low-level effect on the stability of nutrients in milk; however, partial degradation of some nutrients, including water-soluble vitamins B_1_, B_6_, B_12_, and C, some hormones, and unsaturated fatty acids occurs along with the increased degree of heat treatment. The heating duration and manner could be vital for the degradation of nutrients or the level of age gelation in milk when heated with ultra-high temperatures; a treatment of 156 °C/0.116 s showed much less impact on the quality of milk than a treatment of 138 °C/5s, and direct heating showed less impact than indirect heating. Additionally, evaporation operations can harm oxygen-sensitive components, including unsaturated fatty acids and most water- and fat-soluble vitamins. Moreover, higher processing temperatures alter the viscosity of milk. This is particularly important for products requiring the addition of milk (coffee or baked food) or where viscosity is a key quality parameter of end products (yogurt).

Although heat-induced changes on the macro/micro-nutrients and other constituents of milk have been revealed to a certain extent, knowledge of the differences between direct steam heating and indirect steam heating, the characteristics of DSI technology, and the understanding of the bioavailability and bioaccessibility of these components after heat treatment remains insufficient. More in vitro and in vivo studies should be conducted to explore how heating influences milk’s benefits for human health, providing a foundation for optimizing thermal treatment parameters. In addition, the different influences among DSI, DSIJ, and ISH on the food matrix should be further studied to benefit the dairy industry. 

## Figures and Tables

**Figure 1 ijms-25-08670-f001:**
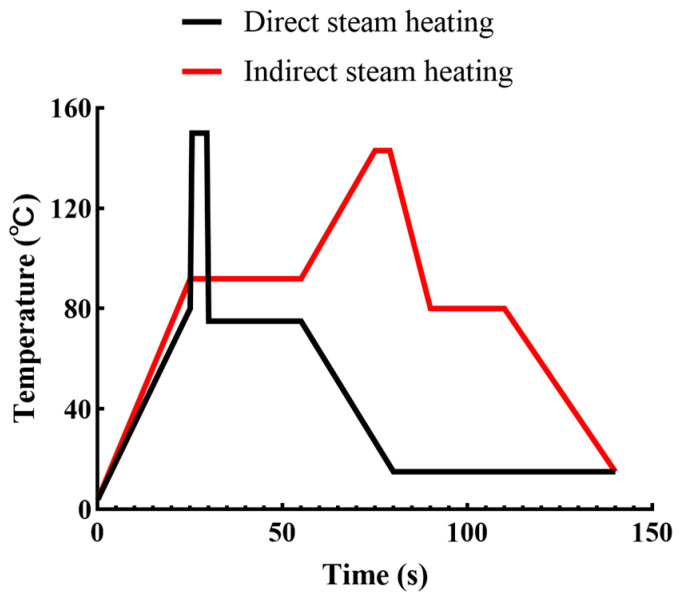
Temperature–time profiles of DSH and ISH.

**Figure 2 ijms-25-08670-f002:**
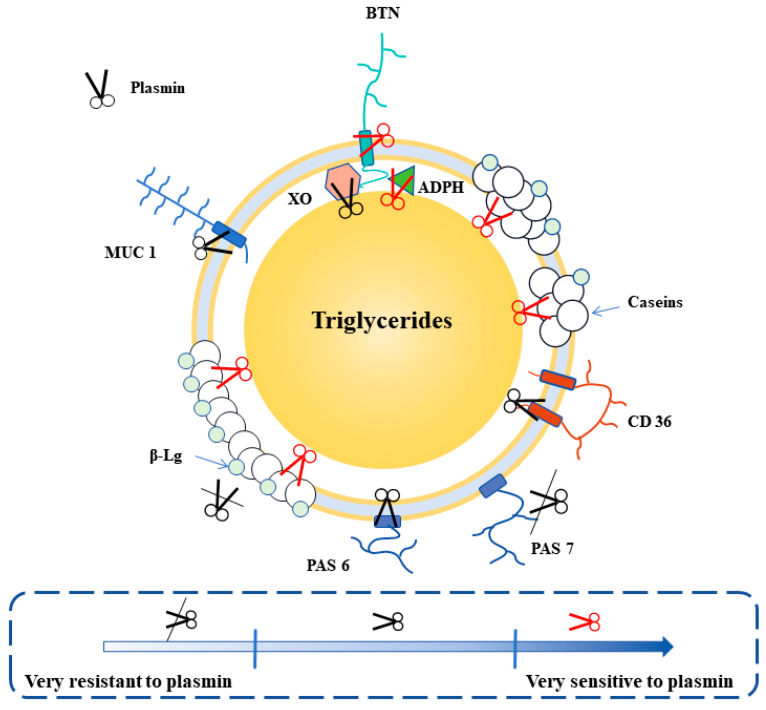
Molecular mechanisms of fat destabilization in UI-DSH whole milk induced by plasmin.

## Data Availability

Data sharing is not applicable to this article.

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
