# Peer review of "The Impact of Thermal Treatment Intensity on Proteins, Fatty Acids, Macro/Micro-Nutrients, Flavor, and Heating Markers of Milk—A Comprehensive Review"

_ijms, 2024, doi:10.3390/ijms25168670_

Round 1

Reviewer 1 Report

Comments and Suggestions for Authors

This is a timely review, considering how much interest there is in raw milk.

L17:  You explored.  The review summarizes.  I recommend that modification.

L20:  That is an incomplete sentence.  

L25:  Again, you tried to understand, but the review summarizes your understanding.  Modify accordingly.

Key words do not need to repeat the word heating.

L33:  Perhaps divide into 2 concepts/2 sentences:  compounds and health benefits.

L36:  MILK consumption has not "grown rapidly over the past few decades" in all countries (particularly in the U.S.  In fact, milk consumption has dropped.  Cheese consumption has increased, however).  Please modify.

L38:  if you just say "microorganisms", it may be more accurate.  pathogenic m/o do not necessarily cause spoilage, but they do cause illness.  

L39:  sterilization (complete elimination of live microorganisms) is NOT recommended/desired.  Pasteurization is adequate in the U.S. and some other countries.

L42:  Please do not use the word sterilization.  It is not accurate.  Modify the 1st sentence in that paragraph and remove all other uses of sterilization unless talking about UHT with aseptic packaging.

The sentence starting at L43 seems a little early.  Perhaps define and make it clear that negative reactions are minimized at HTST and LTLT conditions (define those early in the text and build up to ESL and UHT and potential defects?).

L48:  You have not defined plasmin, so this, too, is too early in the text.

L53:  If initial microbiological numbers are low, some HTST milk has shelf life of ~21 days.  The reference to food waste is superficial.  Either remove it or talk more about it (later in the text?  it seems too early here).

L55:  now you jump into DSH.  I suggest clearly introducing batch pasteurization (LTLT) and continuous (HTST) early, and the benefits of those, before talking about other technologies.

L60:  scalding?  I do not think the INTRODUCTION is a place to talk about alternatives to HTST.  The introduction should introduce the overall manuscript, a little background and preparation for what is to come in the paper.  Growing demand for it?  Where?  Later in the ms, you can certainly bring up this technology, but explain why HTST pasteurization predominates in the industry.

L65:  if DSH is a focus of this ms, it should probably be called out in the title.  Based on the title, I expect a broad overview of thermal treatments used in the industry.  That is not what the introduction suggests I will be reading about.

L70:  Based on the title of this section, I would expect impact of LTLT and HTST heating on milk constituents.  The section focuses on steam.  If this is how the paper is going to go, the title MUST be changed.

L73:  "vacuum cooling"... and removal of steam (excess water) to solids of incoming milk?  Clarify.

L75:  comparable to or compared to?  

L76:  Remove "In addition".  Is "ameliorating" the best word?

Figure 1.  Where did that come from?  Must cite sources (and get permission if published previously).

L85:  Whey proteins (need s)... represent (remove s)

L91:  The explanations in this section should include more specific times and temperatures and detailed chemical changes.  For instance, at what temperature are secondary changes likely?  tertiary changes?

L98:  too vague.  Which whey protein?

L102:  Too vague.  What temperature and time?

L105:  How similar are cow milk, yak milk, buffalo milk, sheep milk and goat milk?  Since there are some differences, please stick to cow milk in this section.  Bring up different species in a different section, later, if you like.

L112:  You have not explained what proteomics or bioinformatics are.  

L116:  more complexes than what?  This sentence is not clear.

L119:  denatured.  not denaturation.  From here on, I will not correct English writing and punctuation because it is slowing me down.  Instead, I will comment only on content.  What is "severe heating"?  

L126:  what "higher temperatures"?

L133:  at what temperature?

What are the consequences of all of these things?  Does it change milk's flavor or functionality very much?  Explain.

L138:  Caseins (add s because there are several types) and account (remove s).

L140:  This is the rough ratio in COW milk.  Clarify.

L147:  at what temperatures do these things happen and what is the importance?  

L153:  explain why they lack tertiary structure?

L157:  not the same as what?  not the same at different temperatures?  say that.

L168:  What is C*?  Chroma?  Explain.

L186:  what are the practical implications of these findings? 

L202:  Here is where you could elaborate on what happens at various temperatures.  "Thermal treatment" is vague.

L208:  times and temperatures are needed

L210:  "loss"?  what does that mean?  Is the milk no longer nutritious.  No, that is not the case, but "loss" makes it sound like it.  Be more clear.

L215:  so is there any homogenizing effect with DSI?

L217:  one polar lipid or polar lipids?

L231 and 234 and 236:  would influence, or influences?

L239:  explain the flavors/aromas noticed by humans for these compounds (not explained in the next sentence).

L241:  I think you mean "cooked aroma"

Explain what the difference is between LLM and ultra-pasteurized milk--or are they the same.  Be consistent with language used.

L251:  what is the difference between DSI and DSIJ?  Explained earlier?  I thought they were the same... 

L253:  you cannot start a sentence with a number (59).  Write it out.

What is the practical finding (meaning to milk consumers) from those results?

What are SFM and SJM?  And what is the practical finding (meaning to milk consumers) from those results?

L282:  IF that is more about cooling than heating, remove.

L283:  what is Pi?

L295:  how "severe"?

L301:  So...  HTST milk has all minerals of raw milk?

L309:  same as above?

L311:  what are "creaturely products"?  

L328:  So...  HTST milk has all vitamins of raw milk?

Address for each vitamin throughout this section, as necessary.

What about furosine, ALP and other "heating markers" for LTLT and HTST milk?

Explain that age-gelation is not an issue in standard fluid milk (LTLT or HTST).

L508:  Ahah!  Here it is.  So take references to other milks out of any earlier sections.  Plus, I would NOT call goat or buffalo milk unconventional.  They are consumed by large populations of people.  

Goat milk does NOT have higher nutritional value.  Please be careful not to say that.  It is deficient in some nutrients, adequate in others, and higher in a few than cow milk, but not better than cow milk.  And it is not because of low b-LG or as1-CN.  Read more about it or remove that.

Frankly, I do not know why you are talking about GM and YM consumption so much.  That is not what this paper is supposed to be about.

Again, YM does not have higher nutritional value.  You cannot just look at protein, fat and dry matter and say it is better than cow milk.  it is more complex than that.

The conclusion should be re-written after major revisions of the paper have been made.  Temperature and time details should be included instead of vague terms like "pasteurization".

Comments on the Quality of English Language

The English language could use some improvements beyond comments I included, above.

Author Response

Thank you a billion for your comprehensive review towards the paper. Your suggestions have made the paper look much better than it was.

please see the attachment for detailed responses.

Reviewer 2 Report

Comments and Suggestions for Authors

This paper reviews a wide range of information on heat treatment of milk and its components.  It lists 156 references, many of which are quite recent. They include several papers by the authors.   Some specific comments follow (line number[s], comment)

23, this statement has little meaning without the conditions of the “ultra-short-time processing” being specified. Further, a process cannot be compared with a substance (milk).

33, 36, 234 and other, use of the terms “etc” and “and so on” are not necessary when following the terms “such as” and “including”

39, I suggest changing “sterilization” to “thermal treatment” as sterilization generally refers to processes such as UHT treatment which render the products (commercially) “sterile”

44-45, please change “exceeded” to “excessive”.  Also, the term “denaturation” should not be applied to lipids

60-61, this statement has little meaning unless the conditions of the DSH treatment are specified. Also, the cited reference (9) does not seem to contain this information. Please check.

63, the word “sterilization” is inappropriate here as DSH is also very useful for use at sub-sterilization conditions, for example, production of ultra-pasteurized or ESL milk

74-75, again, it is meaningless to compare the effects of DSH with pasteurization unless the conditions (time, temperature) of the DSH treatment are specified

93, whey proteins are more sensitive than what?

110, this statement requires the conditions of the DSH treatment to be specified

116, what is meant by “whey proteins with higher molecular weight”?

157, Should “protein size” be “protein particle size”?

162, Please check reference given (No 50) as it is about fish immunostimulants.

165-166, please reword sentence as it currently does not make sense

168, please change “protein size” to “protein particle size” as in cited reference 53

189, I question the use of the word “countless”. I suggest deleting it.

213, Is the statement “DSH has a homogenization effect” true.  DSIJ definitely has but I don’t think DSI (infusion) does.

242, “cooked” is a better term than “cooking”

244, please define “ultra-pasteurized” which is a term used specifically in the USA.

251, this is only meaningful if the conditions used in DSI and DSIJ are specified  

260-264, what are SJM and SFM?

275, delete “concentration”

285-286, I don’t consider heating at 60-80°C to be “intensive”

310-392, I suggest that the first time a vitamin is mentioned, the common name is also given.

Table 1, please also give the common names of the vitamins.

331, I suggest “undermining” be replaced with “reduction”

394-398, It is incorrect to state that furosine is generated in the Maillard reaction.  Furosine is formed from Maillard reaction products by digestion with acid during the chemical analysis of heated milk.

393+, lactulose should be discussed in this section as it is a better heat index than furosine as, unlike furosine, it changes little during storage of milk.

498, what “stabiliser”?          

502, Has it been shown that “it is practicable to screen raw milk with a lower total plasmin activity”?.

607, DSI is not an emerging technology

928-929, Please complete this reference. Is it published?

Comments on the Quality of English Language

Mostly OK but some words used are inappropriate.  

Author Response

(The authors gave the same response as above.)

Round 2

Reviewer 1 Report

Comments and Suggestions for Authors

The manuscript has been improved, but the organization of every section should be improved, such that the impact of LTLT and HTST are clearly explained before more extreme cases.

In particular, please acknowledge clearly throughout the paper and in the abstract that standard HTST (72C, 15 sec) and LTLT (63C, 30 min) pasteurized milk is similar in nutrition and flavor to raw milk.  The paper must make a CLEAR DISTINCTION between these standard heat treatments and excessive heating.  Do this EARLY in the introduction (DEFINE HTST and LTLT in the second paragraph) rather than mentioning excessive heating in the 2nd paragraph. Many of your references deal with higher temperatures than LTLT and HTST, but these are standard US milk treatment temperatures.

L38:  remove "rapidly"

L42:  add back "is an"

L46:  remove "would"

L50:  not simply "heating", but "excessive heating".  STANDARD LTLT and HTST do not cause these problems.  This must be addressed clearly before you get into the more extreme cases, which seem to be the focus of this review. That is why I said you should FIRST discuss HTST and LTLT, then discuss DSH and others.  You basically can re-organize this paragraph.

Section 2 (L85) jumps right into steam injection.  FIRST, you need to talk about LTLT and HTST, as indicated in the last paragraph of the introduction.  Re-organize.  I believe that EVERY section (2, 3, 4, 5, and 6) should START with raw milk, move to LTLT and HTST, then move on to the more extreme cases.

Comments on the Quality of English Language

Some additional English language editing will improve the manuscript.

Author Response

Please see the attachment for detailed responses.

Thank you again for your patience and suggestions.

Reviewer 2 Report

Comments and Suggestions for Authors

The authors have satisfactorily addressed most of the points I raised in my initial report.  However, there are a few points which still require the authors; attention. 

As pointed out in the original review, it is meaningless to compare short-time processing with normal thermal pasteurization unless the temperature of the short-time heating is given.  Please revisit, for example, lines 23-26 of the correct version. A true comparison of heating methods is only valid when the bactericidal effects are the same. It is unrealistic to make comparisons between processes in the UHT temperature range with pasteurization.

 Likewise, when comparing indirect and direct processes, a true comparison of effects is only valid when the processes cause equal bactericidal effects.  This needs to be stated.

The authors continue to state that “Direct steam heating (DSH) [is] an emerging technology applied to dairy industry in recent years” (Lines 65-66 in the corrected version).  They should note that papers comparing direct and indirect UHT heating were published in the 1970s (for example, Franklin, J. G., Underwood, H. M., Perkin, A. G., & Burton, H. (1970). Comparison of milks processed by the direct and indirect methods of ultra-high-temperature sterilization. II. The sporicidal efficiency of an experimental plant for direct and indirect processing. Journal of Dairy Research, 37(2), 219-226”. and DSH methods are described in detail in the 1988 classic book “Burton, H. (1988). Ultra-high Temperature processing of Milk and Milk Products. Elsevier Applied Science”.  Hence DSH is not an “emerging technology”.

Comments on the Quality of English Language

Overall, the English is reasonable but the paper would benefit from some editing 

Author Response

(The authors gave the same response as above.)
